# A town level comprehensive intervention study to reduce salt intake in China: protocol for a cluster randomised controlled trial

Jianwei Xu,[1] Biwei Tang [ORCID] ,[1,2] Min Liu,[1] Yamin Bai,[1] Wei Yan,[3] Xue Zhou,[4] Zhihua Xu [ORCID] ,[5] Jun He,[6] Donghui Jin,[7] Jixin Sun,[8] Yuan Li,[9] Feng J He,[10] Graham A MacGregor,[10] Jing Wu,[1] Puhong Zhang[11]

JX and BT contributed equally.

JX and BT are joint first authors.

For numbered affiliations see end of article.

**Correspondence to**
Prof Jing Wu;
wujingcdc@163.com

Prof Puhong Zhang;
zpuhong@georgeinstitute.org.cn

## ABSTRACT

**Introduction** Salt intake in China (≈12 g/day) is more than twice the upper limit recommended by the WHO (5 g/day). To reduce salt intake, Action on Salt China (ASC) was launched in 2017. As one of four randomised controlled trials (RCTs) in the ASC programme, a comprehensive intervention study was designed to test whether all the components of the interventions adopted by other RCTs are acceptable, scalable and effective when provided to a region in the real world.

**Methods and analysis** Using a cluster RCT design, 2688 participants were selected from 48 towns (clusters) in 12 counties in 6 provinces and assigned to the intervention group or the control group. Randomisation was performed after the baseline survey was completed. Information on salt-related knowledge, attitude and practice (KAP), blood pressure and 24-hour urinary sodium were collected. The intervention includes government engagement, health education and other intervention components targeting restaurants, home cooks and primary school students and their families that have been used in other RCTs. The control group will not receive the intervention. The project will be followed up for 2 years, with the intervention being carried out for the first year only. The primary outcome is salt intake measured by 24-hour urinary sodium excretion after 1 year. The secondary outcomes are the long-lasting effectiveness on salt intake and blood pressure measured by the same method, as well as salt-related KAP and blood pressure at the 1-year and 2-year follow-ups. Process evaluation and health economics analysis will be conducted as well.

**Ethics and dissemination** The study was reviewed and approved by the Institutional Review Board of the National Center for Chronic and Noncommunicable Disease Control and Prevention, the Chinese Center for Disease Control and Prevention, and Queen Mary Research Ethics Committee. Results will be disseminated through presentations, publications and social media.

**Trial registration number** ChiCTR1800018119

### Strengths and limitations of this study

► This study is the first randomisedcontrolled trial of comprehensive intervention in urban salt reduction in China.
► Our comprehensive intervention measures cover different groups of people and different places, and benefit a wide range of people.
► The shortcoming of this study is that the implementation of comprehensive salt reduction interventions requires the strong support and cooperation of local governments and institutions.

hypertension is an independent risk factor for stroke and coronary heart disease, as it accounts for 62% of strokes and 49% of coronary heart disease cases.[1–3] In China, there were approximately 177 million people with hypertension in 2002, and high BP attributed to 2.33 million cardiovascular disease (CVD) deaths in 2005.[4] Hypertension is a 'silent killer', and 75% of hypertensive Chinese individuals are not aware that they have high BP.[5] In China, salt intake is very high, with an average intake rate of 12–14 g/day.[6 7] There is compelling evidence in adults that a modest reduction in salt intake lowers BP and reduces cardiovascular risk.[3 8–11] Salt reduction is one of the most cost-effective ways to prevent CVD in high-income as well as low-income and middle-income countries.[12 13] The daily salt intake that is currently recommended by the WHO is 5 g (the Chinese government recommends 5 g). According to the results of the 2002 National Nutrition Survey, Chinese adults' daily salt intake is twice that of China's recommended dietary guidelines. The WHO has recommended salt reduction as one of the top three prioritised actions to tackle the global non-communicable disease crisis.[14]

## INTRODUCTION

Studies have shown that high salt (sodium) intake is one of the causes of high blood pressure (BP) levels and hypertension, and that

Unlike in diets in developed countries, the major source of salt in the Chinese diet is salt added by the consumers themselves during food preparation.[15 16] In response to the high salt intake, the Action on Salt China (ASC) group was established in 2017 to reduce salt intake by implementing a comprehensive national salt reduction programme.[17] ASC is funded by the National Institute for Health Research of the UK and led by the Queen Mary University of London, The George Institute China, Chinese Center for Disease Control and Prevention (China CDC) and several other key agencies in China. The ASC team has designed a series of programmes, including campaigns for health education and salt reduction in prepackaged food, as well as four cluster randomised controlled trials (RCTs) to develop and test specific interventions targeting different settings or populations. The four RCTs include the (1) application-based intervention study (AIS) targeting schoolchildren and their families[18]; (2) housewife-based intervention study supporting the use of less salt in home cooking; (3) restaurant-based intervention study, targeting salt intake from restaurants and (4) comprehensive intervention study (CIS), which is a scale up study, that is, the study that is reported in this paper.

The CIS was designed to simulate the real world on a large scale, and all the available interventions adopted in the other RCTs will be provided to the local governments of the participating counties. The aims are to evaluate the acceptability of each component of the intervention, the effectiveness and cost-effectiveness of 1 year of the intervention, the long-lasting effectiveness of the intervention 1 year after it is terminated and to provide evidence for a national scale up. This paper describes the study design, implementation and current status of the CIS.

## METHODS AND ANALYSIS
### Study setting and overall design
Considering multiple aspects, such as the geographical location, economic level and dietary habits, this project will be carried out in six provinces, including Hebei, Heilongjiang, Jiangxi, Hunan, Sichuan and Qinghai, which cover the north, south, east, west and central regions in China.

The CIS was designed as a cluster RCT; it was launched in September 2018 and is expected to be completed by the end of 2020. The clusters are 48 towns (called 'streets' in urban areas, but only 'town' is used hereafter for simplification) selected from 12 counties (named 'districts' in urban areas, but only 'county' is used hereafter) across the 6 provinces. Two counties are selected from each province, mainly from rural or suburban areas where people live in a relatively isolated local environment, unlike those in central urban areas. This process may help to minimise contamination among counties and towns. In each county, four towns that have similar population and economic development levels and are not adjacent to each other are selected to prevent imbalances

in potential confounders between the intervention and control groups that may occur due to the small number (48) of clusters and contamination of intervention to the control group. Fifty-six eligible participants were selected from each town and were randomly allocated to receive either the intervention or no intervention and an evaluation during a 2-year follow-up period. The four towns are evenly and randomly allocated into the intervention or control group after the baseline survey has been completed for all four towns within a county. The randomisation methodology is shared in advance with all the investigators at provincial, county and township levels, but the randomisation results are concealed until the centralised randomisation process is complete.

The implementation of the intervention is led by the county investigators with support from local governments, but the intervention is conducted at the town level. To minimise contamination, all the intervention activities must be conducted within the towns participating in the intervention. The intervention will be carried out for the first 12 months. The effectiveness of the intervention will be evaluated after the completion of the 12 months intervention (mid-term assessment) and after another 12 months following the completion of the intervention to examine its long-lasting effectiveness (endpoint assessment).

### Study population and participant recruitment
The target population includes all adult residents in the study sites. In this study, the inclusion criteria for the participants invited for evaluation are those (1) aged 18–75 years; (2) who do not have another family member participating in the study (a maximum of one family member per family is included); (3) who were considered local residents for over 6 months and have no plans to move within the next 24 months and (4) who agreed to sign an informed consent form. The exclusion criteria are (1) pregnant women and those in the lactation period; (2) individuals who currently participate in any other clinical trials; (3) those with severe psychiatric or physical diseases that might impact the intervention and follow-up and (4) individuals who are unable or for whom it is not suitable to collect 24 hours urine due to the following conditions: (a) aconuresis; (b) acute/chronic urinary tract infection, vaginal infection and perianal infection; (c) acute haemorrhagic diseases in the urinary tract, vagina and digestive tract or (d) severe vomiting and diarrhoeic symptoms.

Two-stage sampling is conducted to recruit participants. First, two villages (named committees in urban areas) are randomly selected, and then, 28 eligible participants, that is, 56 participants for each town, are randomly selected from each village. The village and participant selection procedure is conducted by county investigators with a specially designed smartphone application. For the random selection procedure, the names of the villages as well as the names of the residents in the selected villages need to be uploaded to the server through the app, and a centralised randomisation result will be presented

through the app to the county investigators. The reasons for excluding individuals are also recorded through the app.

## Intervention

To promote all intervention components for salt reduction, a multi-section engagement strategy is recommended to local governments at the county, township and village levels. The government agencies and other major stakeholders to be engaged also include local centres for disease control and prevention, women federations, publicity departments, hospitals, schools, restaurants, supermarkets and so on. The county CDCs will lead the intervention at the township level, including mass media publicity and education efforts, interventions for communities, schools and catering units and salt reduction interventions based in primary care institutions. Potential contamination may exist because the intervention is led by investigators at the county level, and residents in the control group may visit people or eat at restaurants in towns participating in the intervention. Selecting towns that are not adjacent to each other and restricting the intervention to the towns participating in the intervention will minimise the amount of contamination. The overall major interventions targeting different populations or settings within the towns participating in the intervention are summarised below.

### Salt reduction publicity within the towns participating in the intervention

Each county will mobilise the whole township society in the intervention group by carrying out various themed publicity activities for salt reduction. These include (1) promoting health knowledge by training and distributing brochures and disseminating core information on salt reduction and by leveraging at least two publicity days or important holidays each year, such as *World Salt Reduction Week* and *National Nutrition Week*; (2) organising mass cultural and publicity activities on 'Salt and Health' at least once a year using local popular forms, such as knowledge contests, family health gastronomy cooking contests or other activities, to create an atmosphere dedicated to the reduction of salt intake; (3) establishing good public environments focused on the reduction of salt intake, such as healthy parks, healthy roads and healthy edible oil stations and (4) promoting salt and health knowledge and skills through posters and social media, such as WeChat public accounts, to broaden the audiences to especially include young and middle-aged groups.

For mass publicity, we encourage leveraging the local culture and customs and using innovative forms and content, and we suggest that the form is diverse, the coverage is broad and the content is updated in a timely manner. Moreover, contamination to the towns in the control group must be minimised by limiting the activities to only the towns participating in the intervention.

### Salt reduction based on community

All communities/villages in the towns participating in the intervention will try to establish a salt reduction environment by hanging posters, introducing slogans and distributing pamphlets in conspicuous places. Community or village family chefs and family members will organise at least one training event on salt reduction every year to improve the knowledge and skills of the attendants or carry out other forms of community-themed activities and distribute salt-restricting spoons and other intervention tools. The salt-restricting spoon is a plastic spoon specially designed to measure salt during cooking. Each spoon holds 2 g of salt, which is convenient for home cooks to count and control the salt used during cooking. For the community/village family chefs and family members, the family salt intake monitoring activities are carried out through the Health Salt WeChat applet or the salt intake evaluation booklet. Audio recording of eight standards, each lasting no more than 2 min, are provided for loudspeaker broadcasting for all villages in the towns participating in the intervention.

### Salt reduction based on school

In all the schools in the towns participating in the intervention, publicity posters should be put up on bulletin boards or school canteens. Information on salt reduction can be broadcast at the school during recess to create a good campus environment focused on salt reduction. Salt and health training activities are conducted at least once a year using opportunities such as centralised teacher training or school parent–teacher meetings. In combination with the local efforts, 'AppSalt activities' (AppSalt is an app-based platform that was designed for promoting salt reduction in primary schools, which is the key intervention in the AIS[18]), 'salt reduction-focused sessions in health education classes', 'showings of salt reduction science animations', 'the production of salt reduction-related handwritten newspapers' and other forms of salt reduction public activities are conducted in schools. In addition, we also need to actively encourage students' parents to use AppSalt by conducting training sessions, organising school WeChat groups, or organising a class on the use of AppSalt.

### Salt reduction based on restaurant

Salt reduction activities will be carried out in restaurants and canteens at workplaces located in the towns participating in the intervention. Information on salt reduction will be shared through posters, table displays and brochures made publicly available to create a restaurant environment conducive to the reduction of salt. In the restaurants and the canteens, the catering chiefs are offered standardised training for at least 1 year on how to reduce the amount of salt and salty sauces used during cooking at least once a year. Consumers are encouraged to order food with less salt from the servers.

### Salt reduction based on primary care service

This project also addresses the community health services under the jurisdiction of the towns (township hospitals) and villages (village clinics) by publicising salt reduction information in the form of posters, accessible brochures or banners in the hospitals and clinics. If possible, these facilities can publicly broadcast salt reduction videos. The county CDCs will provide training at least twice a year for all the healthcare providers under the jurisdiction of the towns participating in the intervention by combining the training with the routine training of basic national public health services. All primary healthcare institutions should hold lectures and informational sessions on salt reduction at least twice a year. Primary care providers should share their salt reduction knowledge and skills during routine outpatient clinic visits to promote the reduction of salt and prevention of hypertension for the visiting patients.

Theoretically, as mentioned above, although some salt reduction activities are mandatory, it is acceptable for the local governments to select some of the intervention tools or materials and add some as they see fit. However, the degree, coverage and cost of the projects must be recorded when they are implemented. No additional interventions on salt reduction will be conducted among the control group in year 1. In year 2, we will try to scale the interventions evenly across the nation through the national CDC system, with no differences in the input or support among the different areas, including those originally randomised into the control group. The hypotheses are that the national scale up has the same impact on the two groups, and the difference in 24-hour urinary sodium excretion at the end of year 2 can reflect the actual long-lasting effectiveness of the intervention conducted in year 1.

### Sample size

We recently completed Shandong salt reduction project with a similar comprehensive intervention, and it showed that the before–after effectiveness of sodium reduction was 37 mmol/day.[19] Considering that a before–after design may overestimate the effectiveness of the intervention, we expect that our intervention, compared with the control condition, will reduce sodium intake by at least 25 mmol/day (1.46 g/day salt) from baseline. The target sample size will have 80% power to detect a change of 25 mmol/day. According to the research design, we randomly selected 2688 eligible participants from 48 towns (56 each) in 6 provinces. Assuming the maximum drop rate is 20% for towns (from 48 to 40) and 10% for participants (from 56 to 50) within the 2-year follow-up period, this sample size and sampling method will have 80% power to detect a difference of 25 mmol/day between the group means, assuming the SD is 85.0 mmol/day and intraclass correlation coefficient (ICC) is 0.080 at the village level, with a two-sided analyses and a significance level of 0.05. The parameter estimations for effectiveness, SD and ICC are also based on those in the Shandong study and are similar to the results of a cluster RCT that was conducted in China.[7]

### Outcomes and outcome assessment

The primary outcome is salt intake measured by 24-hour urinary sodium exertion. The secondary outcomes include the changes in salt-related knowledge, attitude and practice (KAP) and BP.

All outcome assessments, including the questionnaires, physical measurements and 24-hour urine collection, will be carried out at baseline, that is, before randomisation, at the mid-term point (ie, after 1 year of the intervention) and at the endpoint (ie, 2 years from baseline). Both the towns in the intervention group and those in the control group will be assessed in exactly the same way in parallel.

The questionnaire survey will be conducted face-to-face by trained and qualified investigators through a mobile electronic data collection system.[20] According to the content and sequence of the questionnaire, the basic information of the respondents—behavioural risk factors, knowledge, attitudes and behaviours related to the reduction of salt and prevention and control of hypertension—the presence of hypertension and related expenses will be recorded. Physical measurements, including height, weight, waist circumference, BP, heart rate, will be accurately measured by trained researchers using calibrated measuring instruments.

During the 24-hour urine collection, we will ask the respondents to empty their bladder, record the start time, dispense the urine collection equipment, inform the respondents of the urine retention precautions and specify the collection time for the next day. When retrieving the urine collection equipment, we must ask for the last urination time. If the respondent does not remember the time of the last urine collection, then the final urine sample should be collected on site, and the end time should be recorded. If any of the following three problems occur, the urine sample will be deemed unacceptable: (1) the participant forgets to collect or splashes more than 10% of the total amount of urine; (2) the urine is contaminated with blood, stool or other impurities and (3) the participant experiences excessive sweating, diarrhoea or vomiting during collection. If any of the above situations are reported, another 24-hour urine collection should be rescheduled. Finally, the qualified urine is sealed and transported to the laboratory for unified testing. The test included urinary sodium, urinary potassium, creatinine and albumin.

All local staff members participating in the field investigation will be given the appropriate training, which includes tests. Only the staff members who pass the tests can take part in the field work. In addition, they will be compensated accordingly.

## DATA COLLECTION AND ANALYSIS
### Data collection methods

Case report forms will be developed to collect data, including demographic characteristics, salt-related KAP, BP and 24-hour urinary sodium levels, from both the baseline survey and evaluation. In addition to the data

collected at the baseline, participants lost to follow-up and the corresponding reasons must be recorded both at the 12-month follow-up and the long-lasting effectiveness evaluation at 24 months. Electronic data recording technology will be employed for the data collection.

## Data management

The data from this survey were collected through a mobile electronic data acquisition system (mEDC),[20] which was developed by Beijing University of Aeronautics and Astronautics. Relevant data, such as the questionnaire results, physical measurements and data on the urine collection and intervention processes, were collected by the mEDC system.

## Statistical analysis

The effectiveness of the intervention package on the primary and secondary outcomes will be evaluated at 12 months after the intervention and at 24 months after the intervention has been terminated for 1 year. Linear mixed models will be used to model the outcome measures at 12 months (primary analysis), with adjustments for the baseline variable, the participants nested within village units and the villages nested within towns. The group differences will be estimated using least squares estimation. To account for missing data for the continuous outcomes, we will use the likelihood-based random effects model, which uses all available data to provide valid estimates of the intervention effects when data are missing at random. If more than 5% of the data are missing, various missing value imputation methods will be adopted as sensitivity analyses to examine the robustness of the conclusions of the primary analysis.

When evaluating the long-lasting effectiveness of the intervention package at 24 months, the methods mentioned above will be applied, and the major/secondary outcomes at 12 months will be replaced by those at 24 months. We assume that the effects of the scale up will be similar between the intervention and control groups.

SAS (version 9.4) will be used for the statistical analyses. The results will be reported as the mean, SD, SE and 95% CI when appropriate. All analyses are two-sided, and p<0.05 is considered significant.

## Process monitoring and evaluation

To assess the level of fidelity and adaption to the intervention, process monitoring will be carried out throughout the intervention period. Process monitoring includes the evaluation of the indicator systems and methods. For each county, governance, working system and effective intervention, the plan must be ready before the initiation of the intervention. During the implementation period, all the interventions will be recorded and reported on a quarterly basis according to the activity plan. The report includes the content of the activity, the time of the event, the implementer, the participants and other relevant documents, photos and objects. The national and provincial supervision teams will visit the intervention sites quarterly to assess the development of a salt reduction environment, evaluate the posting and placement of posters, brochures and other promotional materials and determine the distribution of materials, lectures and training.

At the end of year 1 and year 2, systematic semistructured interviews will be conducted separately to evaluate the fidelity and acceptability of each component of the intervention. We will adopt an approach consistent with the UK Medical Research Council (MRC) guidelines for the process evaluations of the complex intervention.[21] This approach will enable us to determine whether the intervention is effective and identify the barriers and enablers for the potential scale up. A combination of in-depth interviews will be conducted with study participants, staff members who implemented the intervention at the county, town and village levels and policymakers, and the data collected during the trial will be assessed as well.

## Economic evaluation

Economic evaluations will be carried out from the health sector perspective to compare the comprehensive intervention with usual care, and the evaluations will involve two components: a trial-based economic evaluation and a modelled economic evaluation of long-term costs and outcomes. Intervention costs will include those in delivering the intervention, which shall be recorded by the county investigators, but exclude any research and development costs. The trial-based economic evaluation will be assessed initially in terms of the incremental cost per unit reduction in salt intake and systolic BP. The modelled economic evaluation with discounting will examine the cost, survival, quality of life over lifetime via capturing various health states (including death and CVD events) to estimate incremental cost per life year saved and cost per quality-adjusted life year gained. The transition probabilities across the health states and costs associated with various health states will be based on data in the literature, and the long-term effects of the reduction in salt intake or systolic BP will be identified from the trial findings and/or literature on disease progression. Sensitivity analyses will be carried out to estimate uncertainty about the primary findings associated with different key parameters.

## Project status and timelines

Preparations were made from April 2017 to August 2018. The baseline assessment was initiated from September to December 2018. A total of 2688 eligible participants were successfully recruited from 48 towns (4 of them located in urban areas) of 12 counties (2 of them located in urban areas) and completed the baseline survey. Since two rounds of evaluations of the effects are to be carried out after 12 months and 24 months, a mid-term evaluation will be conducted by the end of 2019, and the final follow-up evaluation will be conducted in December 2019.

## Patient and public involvement

According to the actual situation in the locality, the individuals at the investigation sites will adopt various methods of distributing propaganda and mobilisation efforts and share the motivation and purpose of the investigation with the residents. They will rely on the leadership and support of the local government and grassroots organisations, lead the intervention, make appointments and strive to understand, support and co-operate with the respondents. After the participants are selected, they will sign an informed consent form and then undergo a questionnaire survey, body measurements and urine collection.

## ETHICS AND DISSEMINATION

### Research ethics approval and consent

Written consent will be obtained from all participants, and they will be free to discontinue their participation at any time without an explanation.

### Consent or assent

Consent for participation in the project will be sought both at the cluster level and the individual level. Cluster-level consent of the community will be obtained through a consultation process involving the governments (at the provincial, county and town levels) and village leaders. The project, including the process of randomly assigning communities to the intervention and control groups and the specific interventions, will be explained in a face-to-face meeting. Questions will be answered, and all relevant stakeholder groups will be invited to consult with the members of their group and reflect on the project. Individual consent for participation in the outcome surveys will be obtained from all persons selected in a standard manner via the provision of a participant information sheet, explanation and discussion as required, and the collection of written consent form from those willing to participate.

Written consent will be obtained from all participants, and they will be free to discontinue their participation at any time without an explanation.

### Author affiliations

[1]National Center for Chronic and Noncommunicable Disease Control and Prevention, Chinese Center for Disease Control and Prevention, Beijing, China
[2]School of Public Health, Inner Mongolia Medical University, Hohhot, China
[3]Jiangxi Provincial Center for Disease Control and Prevention, Nanchang, China
[4]Heilongjiang Provincial Center for Disease Control and Prevention, Harbin, China
[5]Qinghai Provincial Center for Disease Control and Prevention, Xining, China
[6]Sichuan Provincial Center for Disease Control and Prevention, Chengdu, China
[7]Hunan Provincial Center for Disease Control and Prevention, Changsha, China
[8]Hebei Provincial Center for Disease Control and Prevention, Shijiazhuang, China
[9]Peking University Health Science Centre, The George Institute for Global Health, Beijing, China
[10]Wolfson Institute of Preventive Medicine, Queen Mary University of London, London, UK
[11]Diabetes Program, The George Institute at Peking University Health Science Center, Beijing, China

**Acknowledgements** The authors would like to thank the street/community residents, catering companies, school teachers, parents of the primary school students, primary care staff and village, town and county leaders for their opinions on the development of the intervention programme.

**Contributors** PZ and JW conceived the project. JX, ML, YB, YL, JW, FJH, GAM and PZ participated in the design and implementation of the project. JX, BT, ML, YB, WY, XZ, ZX, JH, DJ, JS, YL, JW, FJH, GAM and PZ facilitated patient and public involvement and were responsible for setting up the study in each site. BT and JX wrote the first draft of the manuscript, and they contributed equally to this paper. All authors contributed to the refinement of the study protocol and approved the final manuscript.

**Funding** This work is supported by the National Institute of Health Research (NIHR, NIHR Global Health Research Unit Action on Salt China at Queen Mary University of London) using Official Development Assistance (ODA) funding (16/136/77). The views expressed in this publication are those of the author(s) and not necessarily those of the NIHR or the Department of Health and Social Care.

**Disclaimer** The findings of this study will be disseminated through discussions or presentations at selected conferences, peer-reviewed publications and the general media.

**Competing interests** FJH is a member of the Consensus Action on Salt & Health (CASH) group, a non-profit charitable organisation, and its international branch World Action on Salt & Health (WASH) and does not receive any financial support from CASH or WASH. GAM is the Chairman of Blood Pressure UK (BPUK), Chairman of CASH and Chairman of WASH and does not receive any financial support from any of these organisations. BPUK, CASH and WASH are non-profit charitable organisations. All other authors have no competing interests to declare.

**Patient consent for publication** Not required.

**Ethics approval** The study has been reviewed and approved by the Institutional Review Board of the National Center for Chronic and Noncommunicable Disease Control and Prevention, the Chinese Center for Disease Control and Prevention (201807), and the Queen Mary Research Ethics Committee.

**Provenance and peer review** Not commissioned; externally peer reviewed.

### ORCID iDs

Biwei Tang http://orcid.org/0000-0001-6614-9324
Zhihua Xu http://orcid.org/0000-0002-0547-3355

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
