## [Reviewer comments · BMJ Open]

ARTICLE DETAILS

TITLE (PROVISIONAL)	A town level comprehensive intervention study (CIS) to reduce salt intake in China: Protocol for a cluster randomized controlled trial
AUTHORS	Xu, Jianwei; Tang, Biwei; Liu, Min; Bai, Yamin; Yan, Wei; Zhou, Xue; Xu, Zhihua; He, Jun; Jin, Donghui; Sun, Jixin; Li, Yuan; He, Feng; MacGregor, Graham; Wu, Jing; Zhang, Puhong

VERSION 1 – REVIEW

REVIEWER	Christopher R Long Assistant Professor, University of Arkansas for Medical Sciences, USA
REVIEW RETURNED	23-Aug-2019

GENERAL COMMENTS	This protocol describes an ambitious and exciting project. It addresses an important health concern (i.e., sodium consumption), and the study results will be very interesting. However, the protocol includes several inconsistencies in the presentation of the research design that must be addressed and clarified, including confusion around recruitment, randomization, intervention details, and the sample size calculation. There are several areas where the grammar and syntax could be improved. I have listed several concerns below in page number order through the manuscript. Major concerns are marked with three asterisks (“***”). Title ***For clarity, the manuscript title should include the word “protocol” (e.g., “Protocol for a cluster randomized trial”). Abstract: Line 19: Not clear what is meant by “town/street” Line 23: Verb tense suggests towns have already been randomized but baseline survey will take place in the future. Line 28: acronym “KAP” is not defined Line 34: verb tense is confusing here. Why is it past tense and not future (i.e., “The control group will not receive interventions.”) Lines 35: the hypotheses should be directional (i.e., decrease rather than simply “change” from baseline). Page 5 Point 2 is not a complete sentence. Page 6 Line 11: Acronym BP is not defined until line 17 Line 17: Dietary is spelled incorrectly Line 22-23: Sentence beginning with “And healthcare...” is not a complete sentence. Line 48: ASC acronym is not defined. Line 55: To what is “(see above)” referring? Urine collection is not described above here. Page 7 Line 11: It seems odd to combine “town/street” here without
---

	explaining why those two levels of analyses go together. [IS THIS EXPLAINED IN METHOD SECTION?] ***Line 15: The first mention of “cost-effectiveness” is in the purpose of the manuscript. It is not in abstract or introduction before here. A rationale for including this as a purpose of the paper should be described before it is mentioned as a purpose of the manuscript. Cost-effectiveness is not discussed later in the paper, as well. METHOD ***Line 29: 6 provinces X 2 counties X 4 does not equal 40 towns/streets. Is the study actually including 48 towns/streets? This needs to be clarified. ***Line 33 and following: How exactly will the towns/streets be matched? Will they be paired? Are they going to be equivalent across all 40, or will there be 20 matched pairs, with one per pair in the intervention arm of the study? More info is needed. ***Line 53: More information is needed about randomization. Will towns/streets be randomized without respect to the counties and provinces from which they are drawn, or will some effort be made to ensure equal representation within each arm for each county and/or province (e.g., two towns/streets per county will be randomized to the intervention). ***Depending on how randomization is implemented, how will contamination across arms be controlled? If sodium reduction is a national priority, can messaging be targeted only on the intervention streets without spilling over into the control arm streets? Line 55: KAP is not defined. ***Line 55 and 58: How will the “efficacy” 12 month evaluation differ from the “effectiveness” 24 month evaluation? This is not clear. If explained later in the manuscript, the authors may wish to wait to introduce the efficacy vs effectiveness distinction until they are ready to describe it. Page 8 Line 29: This sounds like contamination of messaging into the control group: “salt reduction advocacy will be delivered at national level among the control and intervention groups, the whole society is widely involved...” Will national messaging be received by the control group? Lines 29-30: These should be split into two sentences. ***Line 41 speaks of intervention counties. Is randomization to take place at the county level? This seems to contradict previous mention of randomization at the streets level (page 7 line 53). Line 48: is this section actually “Salt reduction based on counties”? The other sections below that describe the intervention seem to be based on places rather than strategies (e.g., publicity). Page 9 Line 22-26: This is not a complete sentence. Line 41 shifts into past tense. ***In general the description of the intervention needs to be more specific, if possible, to characterize the dosage that is delivered at each level of intervention. Can you say how many posters in how many restaurants per town/street or county? How many activities per school? How many chat messages per day or week? And so on across each activity. Perhaps a table of activities might help, summarizing and quantifying them as specifically as possible (how much dosage within how many sites per town or county or whatever applicable level of intervention). If some intervention activity occurs in every restaurant or every school, that would also be important to note. Ideally, the intervention would be described in enough detail that someone else could implement a similar version in a different study.
--	--

	Page 11 ***Line 10: If the control arm receives the intervention in Year 2, this seems like it would contaminate the 24 month follow up in the control arm. That is, the control arm’s 24 month data would then be affected by having received the intervention. ***Line 16: This says no intervention will be conducted in the control group. This seems to contradict the paragraph above, which says the control group will receive the intervention in Year 2. ***Line 21: The sample size calculation suggests that it is based on 40 clusters, suggesting that towns/streets are the unit of randomization. That seems to contradict the earlier discussion of “intervention counties.” ***Line 21: Why is 25 mmo/L the target for reduction? Please explain why this number was selected. Line 25: Why is the standard deviation set at 85? Why is the within-pair coefficient of variation set at 0.08? Please explain why these numbers were selected. ***Line 29: Why 2500 individuals? 40 clusters times 50 people = 2000 individuals, or 1000 per arm. ***Based on the sample size calculation as described here, this appears that the study is a cross-sectional design. Given that a longitudinal design would be more powerful, could you provide a sentence explaining what led the research team to select a cross-sectional approach relative to trying to track participants over the 24 months of the study? ***There should be a section on participant recruitment for the data collection described near here rather than at the very end of the protocol. Will they be selected by government/village leaders? Will they be recruited from the face-to-face meetings? How exactly will people be recruited? Will they be compensated? Line 48: Here the language shifts into past tense again. Page 12 Line 23: Please clarify what is meant by “whether the urine is qualified.” Line 38: Please define the acronym CRF ***Line 43: The discussion of “participants lost” at 12 and 24 months suggests a longitudinal (not cross-sectional) design. If participants are being tracked across the 24 months, then this should be clarified throughout the method section and should affect the sample size calculations. If this is a longitudinal design, retention activities and expected attrition rates should also be addressed. (Missing data appear to be included in the analysis plan, so this does seem to be a longitudinal design.) Page 13 ***Line 9: If only one participant per family is allowed to participate, then it does not seem necessary to nest people within family units for analysis. Please explain or revise this choice. ***Line 9: The sample size calculation was based on a t-test, but the analyses are mixed linear modeling (and appear longitudinal). The sample size calculation probably needs to be revised. ***Line 54: When the process evaluation team provides its quarterly report, is there some procedure in place to ensure that the study team can act on any problems identified by the process evaluation team? It should be described how these quarterly reports will be used by the study team. The project status and timeline should perhaps be presented earlier in the manuscript to explain why some activities are presented in the past tense and others are presented in future tense.
--	--

REVIEWER	Dr Yannan Jiang
----------	-----------------

	The University of Auckland
REVIEW RETURNED	31-Aug-2019

GENERAL COMMENTS	This is a large cluster randomised trial conducted in China across six different provinces with 40 towns/streets (the clusters) randomly allocated to either intervention or control groups. A baseline survey was carried out before randomisation among 2500 participants. Two follow up evaluations will be completed at 12 and 24 months. Although the authors have included most details on the design and analysis of the trial, substantial revision is still needed to meet the standard requirements for a study protocol. Please follow the SPIRIT 2013 statement and checklist with recommended items to be included in a clinical trial protocol. Please also carefully check each sentence and use professional English wording and grammar. What clusters are used in a cluster randomised trial is critical to the trial design. With multistage sampling units at the provincial level, it would be helpful to provide more information on the hierarchical levels of different units, e.g. community/village, towns/streets, county/district etc. The authors mentioned that four towns/streets were selected from each county/district, and two counties/districts were selected from each province. This gives a total of 48 towns/streets across six provinces, not 40. It is also confusing that, in the sample size calculation, a sample of 20 cluster pairs (40 in total) with 50 individuals per cluster are targeted. This gives a total of 2000 individuals required for each wave of evaluation, not 2500. Please provide full details on the definition of clusters and proposed cluster sizes at baseline, 12 months and 24 months. For the participants who completed the baseline survey, will the same individuals be followed up at 12 and 24 months for outcome assessments? This needs to be stated clearly as it is related to the sample size calculation and statistical analysis. For a two years study, the intervention strategies will be made available to the general public after one year, including the control group. If this is the case, how will the long-term effectiveness of the intervention package be evaluated at 24 months, when both groups have received the intervention in the past year? The primary outcome is defined as the decrease of 25 mmol/L in the 24-hour urinary sodium level from the baseline. How much the urinary sodium level will decrease at the end of intervention is the research hypothesis, not the outcome measure itself. The linear mixed models will include group, time and their interaction where the time includes both baseline and the end of the trial. The 24 months assessments will be evaluated separately using similar models. Note that the baseline outcome value is normally treated as a baseline confounder and adjusted in the model as a covariate. Only those that are measured post randomisation are considered as study outcomes, which are likely to change after intervention. A better modelling approach is to include both outcome measures at 12 and 24 months and adjust for the baseline value. The group difference will be estimated at each time point using the interaction term. The random effects include both the clusters at the level of randomisation and repeated measures at the participant level. The authors stated that they will adjust for the stratification variables at randomisation and potential confounding variables. They should be pre-specified, as no stratification was mentioned in
--

	randomisation. In addition, as one of the study objectives, how will the cost-effectiveness analysis be conducted at 12 and 24 months?
--	--

VERSION 1 – AUTHOR RESPONSE

Reviewer: #1

This protocol describes an ambitious and exciting project. It addresses an important health concern (i.e., sodium consumption), and the study results will be very interesting. However, the protocol includes several inconsistencies in the presentation of the research design that must be addressed and clarified, including confusion around recruitment, randomization, intervention details, and the sample size calculation. There are several areas where the grammar and syntax could be improved. I have listed several concerns below in page number order through the manuscript. Major concerns are marked with three asterisks (“***”).

Title

1. ***For clarity, the manuscript title should include the word “protocol” (e.g., “Protocol for a cluster randomized trial”).

Response: Thank you for the advice and we have added the word "protocol" in the title and changed the title to “A town level comprehensive intervention study (CIS) to reduce salt intake in Chinese residents: Protocol of a cluster randomized controlled trial”. In addition, this study (CIS) is one of the four RCTs within ASC (Action on Salt China) which is funded by National Institute for Health Research (NIHR) of UK. To make the relationship clear, we have added several sentences in the Introduction part.

Abstract:

2. Line 19: Not clear what is meant by “town/street”.

Response: In China, the administrative units, although at same level, usually have different names in urban and rural areas. In our study, the study sites (clusters) are towns under a county in rural area, and streets under a district in urban area. County and district are the same level under a province or a city. In our study, each province selects 2 project counties mainly from rural and suburb areas where the livings of local residents are more independent of surroundings when compared with those in central urban area. This may help to minimize contamination among counties and towns. This has been explained in the Overall design part.

3. Line 23: Verb tense suggests towns have already been randomized but baseline survey will take place in the future.

Response: The randomization can only be conducted when baseline survey have been completed. We have rewritten this section as Study Setting and Overall Design under the METHODS AND ANALYSIS.

4. Line 28: acronym “KAP” is not defined

Response: Many thanks. We have defined it at its first appearance in the abstract.

5. Line 34: verb tense is confusing here. Why is it past tense and not future (i.e., “The control group will not receive interventions.”)

Response: Our plan has been implemented in 2018, so the past tense is used here.

6. Lines 35: the hypotheses should be directional (i.e., decrease rather than simply “change” from baseline).

Response: We have removed the word "change" because we just to describe the primary/secondary outcomes, rather than the hypotheses. The hypotheses will be clarified in the main text, especially in the Sample size part and Data analysis part.

Page 5

7. Point 2 is not a complete sentence.

Response: Thank you for the advice and we have deleted this sentence and reorganized Strengths and limitations of this study in the article.

Page 6

8.Line 11: Acronym BP is not defined until line 17

Response: Thanks, wWe have defined BP at its first appearance on line 14 in page 6.

9.Line 17: Dietary is spelled incorrectly

Response: We have corrected the wrong word "Dirtary" to "Dietary".

10.Line 22-23: Sentence beginning with "And healthcare..." is not a complete sentence.

Response: We don't think this is needed here, so we have removed it.

11.Line 48: ASC acronym is not defined.

Response: We have added here the definition of ASC, "Action on Salt China".

12.Line 55: To what is "(see above)" referring? Urine collection is not described above here.

Response: Here is a writing error, we have removed "see above".

Page 7

13.Line 11: It seems odd to combine "town/street" here without explaining why those two levels of analyses go together. [IS THIS EXPLAINED IN METHOD SECTION?]

Response: Please refer to the response to comment 2. In our study, there are 48 towns (named street in urban areas, but only "town" is used for simplification) selected from 12 counties (named district in urban areas, but only "county" is used). In fact, there are 10 counties and 2 district among the counties we choose.

14.***Line 15: The first mention of "cost-effectiveness" is in the purpose of the manuscript. It is not in abstract or introduction before here. A rationale for including this as a purpose of the paper should be described before it is mentioned as a purpose of the manuscript. Cost-effectiveness is not discussed later in the paper, as well.

Response: We have mentioned or implicated "process evaluation and health economics analysis" in the Abstract and Introduction parts, and added a detailed description of cost-effectiveness analysis in the middle of DATA COLLECTION AND ANALYSIS.

METHOD

15.***Line 29: 6 provinces X 2 counties X 4 does not equal 40 towns/streets. Is the study actually including 48 towns/streets? This needs to be clarified.

Response: Thanks for reminding us for the mistake. This mistake has existed in several versions of the protocol. Fortunately, the total sample size is correct. We have changed the number of towns/streets in the text to 48. To clarify the calculation, I copy the sentence here: "we randomly select 2688 eligible participants from 48 towns (56 each) in 6 provinces. Assuming the maximum drop rate is 20% for towns (from 48 to 40) and 10% for participants (from 56 to 50) within the two years follow-up, this sample size and sampling method would have a 80% power to detect a difference of 25 mmol/d between the group means assuming the standard deviation is 85.0 mmol/d and intraclass correlation coefficient (ICC) is 0.080, with a two-sided analyses and a significance level of 0.05."

16.***Line 33 and following: How exactly will the towns/streets be matched? Will they be paired? Are they going to be equivalent across all 40, or will there be 20 matched pairs, with one per pair in the intervention arm of the study? More info is needed.

Response: As explained in responses to comment 2 and 13, towns (the name in rural areas) and streets (the name in urban areas) are the same level administrative regions under a county (the name in rural area) or a district (the name in urban area). For reason of simplification, we only use the names in rural areas in the MS. Another reason we use the rural names is that the participating provinces mainly select towns from rural or suburb areas where the livings of local residents are more independent of surroundings when compared with those in central urban area. This may help to minimize the contamination among counties and towns. The sampling and randomization framework can be summarized as: 6 provinces * 2 counties/province * 4 towns/county * 56 pts/town. The towns are the clusters of randomization for treatment (intervention or control). These have been clarified in much detail in the text.

17.***Line 53: More information is needed about randomization. Will towns/streets be randomized without respect to the counties and provinces from which they are drawn, or will some effort be made to ensure equal representation within each arm for each county and/or province (e.g., two towns/streets per county will be randomized to the intervention).

Response: Sorry for making you confused. The randomization will take place among the four towns under a county when the baseline survey for the four towns has been completed. This will help us to balance the potential confounders at town level and above, such as different policies or regulations among towns, counties and provinces. However, we admit the representative to the urban areas might not be very good due to most of the study sites come from rural areas.

18.***Depending on how randomization is implemented, how will contamination across arms be controlled? If sodium reduction is a national priority, can messaging be targeted only on the intervention streets without spilling over into the control arm streets?

Response: In the manuscript, we have explained this part. In each county, 4 towns, with similar population and economic development level, and not adjacent to each other, are selected with the purpose of avoiding imbalance on potential confounders between intervention and control groups due to the small number (48) of clusters and contamination of intervention to control group. At the same time, contamination to the control towns must be avoided by limiting the activities within the intervention towns.

19.Line 55: KAP is not defined.

Response: We have defined it at its first appearance in the text on line 49 in page 12.

20.***Line 55 and 58: How will the “efficacy” 12 month evaluation differ from the “effectiveness” 24 month evaluation? This is not clear. If explained later in the manuscript, the authors may wish to wait to introduce the efficacy vs effectiveness distinction until they are ready to describe it.

Response: Our study is to evaluate the effectiveness of salt reduction strategies in real world, the efficacy is not suitable, so we replaced efficacy with effectiveness throughout the manuscript.

Page 8

21.Line 29: This sounds like contamination of messaging into the control group: “salt reduction advocacy will be delivered at national level among the control and intervention groups, the whole society is widely involved...” Will national messaging be received by the control group?

Response: This is our expectation during scale-up in the future. In the manuscript, we have modified this part, and all of our interventions are within the intervention towns. Efforts to avoid contamination have also been mentioned in several parts of the manuscript.

22.Lines 29-30: These should be split into two sentences.

Response: Thank you for the advice and we have modified it as follows “In order to promote all intervention components for salt reduction, a multi-section engagement strategy is recommended to local governments at county, township and village levels. The government agencies and other major stakeholders to be engaged also include local centers for disease control and prevention (CDC), women federations, propaganda centers, hospitals, schools, restaurants, supermarkets, etc..”.

23.***Line 41 speaks of intervention counties. Is randomization to take place at the county level? This seems to contradict previous mention of randomization at the streets level (page 7 line 53).

Response: Sorry for our carelessness. As mentioned before and in the main text, the intervention will be implemented within towns in intervention group, but the intervention will be led by county CDCs. These have been confirmed carefully together with measures avoiding contamination throughout the manuscript.

24.Line 48: is this section actually “Salt reduction based on counties”? The other sections below that describe the intervention seem to be based on places rather than strategies (e.g., publicity).

Response: Yes, as mentioned above, the intervention will be led by the county investigators with the support of county governments. Multi-section engagement is the key strategy and the basis. The overall health promotion or publicity for salt reduction, and the major interventions targeting different populations or settings within the intervention towns are also summarized in the manuscript. This have been made clear in the Intervention part.

Page 9

25.Line 22-26: This is not a complete sentence.

Response: We have deleted this sentence. And changed this paragraph to "In the process of mass publicity, we encourage to leverage the local culture and customs, use innovative forms and content, and encourage that the form is diverse, the coverage is wide, and the content is updated in a timely manner. At the same time, contamination to the control towns must be avoided by limiting the activities within the intervention towns."

26.Line 41 shifts into past tense.

Response: We changed "were carried" to "carry".

27.***In general the description of the intervention needs to be more specific, if possible, to characterize the dosage that is delivered at each level of intervention. Can you say how many posters in how many restaurants per town/street or county? How many activities per school? How many chat messages per day or week? And so on across each activity. Perhaps a table of activities might help, summarizing and quantifying them as specifically as possible (how much dosage within how many sites per town or county or whatever applicable level of intervention). If some intervention activity occurs in every restaurant or every school, that would also be important to note. Ideally, the intervention would be described in enough detail that someone else could implement a similar version in a different study.

Response: Thank you for your suggestion. We have supplemented this part and quantified some of the interventions. In salt reduction publicity, we provide audios for loudspeaker broadcasting for all villages in the intervention towns, including eight standards, no more than 2 minutes. We do salt and health training at least once a year in schools and restaurants. And the county CDCs carry out training at least twice a year for all the care providers under the jurisdiction of the intervention towns. However, considering this is a multi-section engaged complex intervention, it is acceptable for the local governments to select some of the intervention tools or materials and add some if they think reasonable. But the degree, coverage and cost will be recorded during the project implementation. This has been mentioned at the end of Intervention part.

Page 11

28.***Line 10: If the control arm receives the intervention in Year 2, this seems like it would contaminate the 24 month follow up in the control arm. That is, the control arm's 24 month data would then be affected by having received the intervention.

Response: As mentioned in METHOD AND ANALYSIS, the effectiveness of intervention will be evaluated right after completion of the 12 months intervention (mid-term assessment) and after another 12 months follow-up to find its long-lasting effectiveness (endpoint assessment). But you are right, the control group will be "contaminated" during the second 12 months due to the national scale-up. Our hypotheses is that the national scale-up should have the same impact to the intervention group. That means if the scale-up has the same impact to both groups, the difference on salt reduction at the end of the second 12 months will make the "contamination" of scale-up disappeared and only leave the long-lasting effectiveness. We agree, there may be a lot of uncertainties and uncontrolled actions within the two groups during the last year, but we have no choice, because we and also the nation cannot wait for another year to promote salt reduction in China. We have add the hypotheses and theory at the end of the Intervention part, but did not explain why we do not delay the scale-up for one year. It need a lot explanation. We will clarify this in the future long-lasting effectiveness of the intervention. Hope you are satisfied with this.

29.***Line 16: This says no intervention will be conducted in the control group. This seems to contradict the paragraph above, which says the control group will receive the intervention in Year 2.

Response: Please refer to the explanation right above. To summarized, as written in the main text, "No additional intervention on salt reduction will be conducted among the control group in Year 1. In Year 2, intervention strategies will be made available to the general public, including those originally randomized into the control group. The hypotheses is that the national scale-up has the same impact

to the two groups, and the difference on 24-hour urinary sodium excretion at the end of year 2 can reflex the pure long-lasting effectiveness of the intervention conducted in year 1. ”

30.***Line 21: The sample size calculation suggests that it is based on 40 clusters, suggesting that towns/streets are the unit of randomization. That seems to contradict the earlier discussion of “intervention counties.”

Response: We have checked throughout the manuscript and corrected all the mistakes. The correct point is that the town is the randomized clusters and the unit to receive intervention or control, although the implementation is conducted by county investigators with the support of local governments.

31.***Line 21: Why is 25 mmol/L the target for reduction? Please explain why this number was selected.

Response: 25mmol/L is the expected minimum effectiveness. This is based on our unpublished result from SMASH study conducted by China CDC, in which, the intervention is also a region-covered complex intervention, and the before-after sodium reduction was 37 mmol/l after one year intervention. This has been reported in many conferences. [Chen X, Guo X, Ma J, et al. Urinary sodium or potassium excretion and blood pressure in adults of Shandong province, China: preliminary results of the SMASH project. *J Am Soc Hypertens* 2015, 9(10): 754-762.]

32.Line 25: Why is the standard deviation set at 85? Why is the within-pair coefficient of variation set at 0.08? Please explain why these numbers were selected.

Response: The target sodium reduction (25 mmol/l), deviation and ICC (0.08) are all based on the unpublished results from SMASH study as mentioned above. These are also similar to those found in School EduSalt which was conducted by The George Institute, China. [He FJ, Wu Y, Feng XX, Ma J, Ma Y, Wang H, Zhang J, Yuan J, Lin CP, Nowson C, MacGregor GA. School based education programme to reduce salt intake in children and their families (School-EduSalt): cluster randomised controlled trial. *BMJ*. 2015 Mar 18;350:h770. doi: 10.1136/bmj.h770.] The baseline of Shandong study and the result of School EduSalt have been cited in the part of sample size calculation part.

33.***Line 29: Why 2500 individuals? 40 clusters times 50 people = 2000 individuals, or 1000 per arm.

Response: The reason is “We randomly select 2688 eligible participants from 48 towns (56 each) in 6 provinces. Assuming the maximum drop rate is 20% for towns (from 48 to 40) and 10% for participants (from 56 to 50) within the two years follow-up, this sample size and sampling method would have a 80% power to detect a difference of 25 mmol/d between the group means assuming the standard deviation is 85.0 mmol/d and intraclass correlation coefficient (ICC) is 0.080, with a two-sided analyses and a significance level of 0.05.”

34.***Based on the sample size calculation as described here, this appears that the study is a cross-sectional design. Given that a longitudinal design would be more powerful, could you provide a sentence explaining what led the research team to select a cross-sectional approach relative to trying to track participants over the 24 months of the study?

Response: As already explained elsewhere in the manuscript, this is purely a cluster RCT although the intervention is complex and adaption to the intervention is acceptable.

35.***There should be a section on participant recruitment for the data collection described near here rather than at the very end of the protocol. Will they be selected by government/village leaders? Will they be recruited from the face-to-face meetings? How exactly will people be recruited? Will they be compensated?

Response: We add the participant recruitment in the Study population and participant recruitment part. Here is a copy: “A two-stage sampling is conducted to recruit eligible participants. Firstly, two villages (named committees in urban areas) are randomly selected, and then 28 eligible participants are randomly selected from each village, i.e. 56 participants for each town. The procedure of village and participant selection is conducted by county investigators with the support of a specially designed smartphone application. To fulfil the random selection, the names of villages as well as the names of residents in the selected villages need to be uploaded to the server through the app, and a centralized randomization result will be presented through the app to the county investigators. The reasons why not eligible for some residents are also recorded through the app. ”

The recruitment and surveys will be compensated according to the input of effort as labour fee.

36.Line 48: Here the language shifts into past tense again.

Response: We changed "were carried" to "carry".

Page 12

37.Line 23: Please clarify what is meant by "whether the urine is qualified."

Response: We meant the qualified urine as no following situations reported by participants: (1) forget to collect or splash urine more than 10% of the total; (2) urine is contaminated with blood, stool or other impurities; or (3) excessive sweating, diarrhea or vomiting during collection. As the wording of "qualified urine" may be inappropriate, we replace "qualified" with "acceptable".

38.Line 38: Please define the acronym CRF

Response: We have supplemented the definition of CRF, that is Case report forms.

39.***Line 43: The discussion of "participants lost" at 12 and 24 months suggests a longitudinal (not cross-sectional) design. If participants are being tracked across the 24 months, then this should be clarified throughout the method section and should affect the sample size calculations. If this is a longitudinal design, retention activities and expected attrition rates should also be addressed.

(Missing data appear to be included in the analysis plan, so this does seem to be a longitudinal design.)

Response: Yes, this is a cluster RCT with two years follow-up. The potential drop rates for cluster (20% for town, due to it is a complex intervention and the local governments may retreat if they are too occupied by other things) and participants (10%, a low rate due to stable local resident) are both considered in the Sample size calculation part.

Page 13

40.***Line 9: If only one participant per family is allowed to participate, then it does not seem necessary to nest people within family units for analysis. Please explain or revise this choice.

Response: Yes, you are correct. Many thanks. We have removed family as a nested variable.

41.***Line 9: The sample size calculation was based on a t-test, but the analyses are mixed linear modeling (and appear longitudinal). The sample size calculation probably needs to be revised.

Response: We have modified the sample size calculation, please refer to 33.

42.***Line 54: When the process evaluation team provides its quarterly report, is there some procedure in place to ensure that the study team can act on any problems identified by the process evaluation team? It should be described how these quarterly reports will be used by the study team.

Response: Thank you for your suggestion. We explained it in detail in the manuscript. In order to ensure to supervise the fidelity and adaption to the intervention, process monitoring will be carried out throughout the intervention period of time. And according to the quarterly records and reports, the monitoring team will visit the intervention site on a quarterly basis to check the project implementation. In addition, at the end of year 1 and year 2, a systematic semi-structured interviews will be conducted separately to evaluate whether, why and how the specific interventions work in different settings, with the purpose of promoting the scale-up for effective salt reduction strategies and measures in China and worldwide. These have been added in the Process monitoring and process evaluation part.

43.The project status and timeline should perhaps be presented earlier in the manuscript to explain why some activities are presented in the past tense and others are presented in future tense.

Response: Thank you for the advice and we have added it at the very beginning of Overall Design and at the Project Status and Timelines part.

Reviewer: #2

This is a large cluster randomised trial conducted in China across six different provinces with 40 towns/streets (the clusters) randomly allocated to either intervention or control groups. A baseline survey was carried out before randomisation among 2500 participants. Two follow up evaluations will be completed at 12 and 24 months.

Although the authors have included most details on the design and analysis of the trial, substantial

revision is still needed to meet the standard requirements for a study protocol. Please follow the SPIRIT 2013 statement and checklist with recommended items to be included in a clinical trial protocol. Please also carefully check each sentence and use professional English wording and grammar.

1. What clusters are used in a cluster randomised trial is critical to the trial design. With multistage sampling units at the provincial level, it would be helpful to provide more information on the hierarchical levels of different units, e.g. community/village, towns/streets, county/district etc. The authors mentioned that four towns/streets were selected from each county/district, and two counties/districts were selected from each province. This gives a total of 48 towns/streets across six provinces, not 40. It is also confusing that, in the sample size calculation, a sample of 20 cluster pairs (40 in total) with 50 individuals per cluster are targeted. This gives a total of 2000 individuals required for each wave of evaluation, not 2500. Please provide full details on the definition of clusters and proposed cluster sizes at baseline, 12 months and 24 months. For the participants who completed the baseline survey, will the same individuals be followed up at 12 and 24 months for outcome assessments? This needs to be stated clearly as it is related to the sample size calculation and statistical analysis.

Response : We have corrected the data of the sampling unit. We have selected 6 provinces, selected two counties in each province, and selected 4 towns in each county, so we finally got 48 towns. We randomly selected 2 villages or neighborhood committees in each town, randomly selected 28 eligible participants in each village. Finally, we investigated 2,688 members. We then followed these two years for follow-up, and they were all in three surveys. The same concerns have been also raised by other reviewers. Please find detailed answers there or in the revised manuscript in the Methodology part.

2. For a two years study, the intervention strategies will be made available to the general public after one year, including the control group. If this is the case, how will the long-term effectiveness of the intervention package be evaluated at 24 months, when both groups have received the intervention in the past year?

Response : This question is also raised by other reviewers. As mentioned in METHOD AND ANALYSIS, the effectiveness of intervention will be evaluated right after completion of the 12 months intervention (mid-term assessment) and after another 12 months follow-up to find its long-lasting effectiveness (endpoint assessment). But you are right, the control group will be "contaminated" during the second 12 months due to the national scale-up. Our hypotheses is that the national scale-up should have the same impact to the intervention group. That means if the scale-up has the same impact to both groups, the difference on salt reduction at the end of the second 12 months will make the "contamination" of scale-up disappeared and only leave the long-lasting effectiveness. We agree, there may be a lot of uncertainties and uncontrolled actions within the two groups during the last year, but we have no choice, because we and also the nation cannot wait for another year to promote salt reduction in China. We have add the hypotheses and theory at the end of the Intervention part, but did not explain why we do not delay the scale-up for one year. It need a lot explanation. We will clarify this in the future long-lasting effectiveness of the intervention. Hope you are satisfied with this.

3. The primary outcome is defined as the decrease of 25 mmol/L in the 24-hour urinary sodium level from the baseline. How much the urinary sodium level will decrease at the end of intervention is the research hypothesis, not the outcome measure itself. The linear mixed models will include group, time and their interaction where the time includes both baseline and the end of the trial. The 24 months assessments will be evaluated separately using similar models. Note that the baseline outcome value is normally treated as a baseline confounder and adjusted in the model as a covariate. Only those that are measured post randomisation are considered as study outcomes, which are likely to change after intervention. A better modelling approach is to include both outcome measures at 12 and 24 months and adjust for the baseline value. The group difference will be estimated at each time point using the interaction term. The random effects include both the clusters at the level of randomisation and repeated measures at the participant level.

Response : As your suggestion, we have redefined the outcomes as "The primary outcome is sodium

intake measured by 24-hour urinary sodium excretion. The secondary outcomes include the change in salt-related knowledge, attitude and practice (KAP), and blood pressure.” Also we have adopted your suggestion to model both outcome measures at 12 months (primary analysis) and 24 months adjusting for the baseline value, with participants nested within village units.

4.The authors stated that they will adjust for the stratification variables at randomisation and potential confounding variables. They should be pre-specified, as no stratification was mentioned in randomisation. In addition, as one of the study objectives, how will the cost-effectiveness analysis be conducted at 12 and 24 months?

Response: Considering the RCT design and randomization among the towns within a county, we hope all the confounders should have been balanced and only the baseline value (same as the analysed outcome) is controlled. We give a detailed description for a cost-effectiveness analysis in the Economic evaluation under the METHODS AND ANALYSIS.

VERSION 2 – REVIEW

REVIEWER	Christopher R. Long University of Arkansas for Medical Sciences
REVIEW RETURNED	24-Oct-2019

GENERAL COMMENTS	This is an ambitious and important project, and it is clear that the authors have spent time carefully revising this protocol based on reviewer comments. The introduction is well done--the explanation about the four RCTs is particularly helpful. There are still some minor areas where grammar and usage could be improved--I have listed a few in my comments, but I did not attempt to capture every sentence that could be improved. There a small number of areas that require more clarification. These major concerns are marked with three asterisks (***) . I look forward to reading about the results of this important and potentially influential study. ABSTRACT: "house cook" could be "home cooks" "The secondary outcomes is the..." should read as the "The secondary outcomes are the ..." STRENGTHS & LIMITATIONS: In point #4, "and understand" could be "to understand" Point #5 should be be divided into two sentences. Also "under the same county" could be "within the same county." INTRODUCTION: Pg 6 Line 29: The abbreviation LMIC is not used in the remainder of the manuscript and therefore is not necessary to introduce. METHODS AND ANALYSIS: Pg 7 line 46: "The clusters are 48 towns (named street in urban areas,... " could be clarified to "The clusters are 48 towns (called "streets" in urban areas,... " Pg 7 line 52: In the phrase "where the livings of local residents", the word livings is confusing and could be replaced with a different word. Pg 8 line 17: "the result of randomization keeps concealed " could be "the result of randomization is concealed" Pg 8 line 26: "To avoid contamination," may be more precisely stated as "To minimize contamination," Pg 8 line 26: To minimize the introduction of multiple terms
---

	describing the same concept, is this "township or the cluster level" more precisely described as "the town level"? Pg 9 line 21: "The reasons why not eligible for some residents" could be "The reasons why specific residents are ineligible" INTERVENTION: ***If people are randomized at the town level, but interventions will be carried out by county level staff, it is mentioned that any county level interventions could contaminate the control variables. Are all intervention activities implemented at the town or village level? It is mentioned under the "Salt reduction publicity section" but this perhaps should be emphasized clearly in the beginning of the intervention section. Pg 9 line 35: The term "propaganda centers" is likely to have a generally negative connotation for many English-speaking non-Chinese readers. There may be a more precise term to describe these locations for readers not from China. Throughout the manuscript, whenever contamination is mentioned, it may be more accurate to say that contamination will be "minimized" rather than "avoided" Pg 10 line 26: It is not clear what is meant by "salt limiting spoons" Pg 11 line 16: Does "restaurants with a certain scale" mean "restaurants who serve a large enough number of clients," "restaurants who have a large enough number of locations," or something else? Pg 12 line 10: ***Why is the hypothesis that the national level scale up will have the same impact across both groups? A little more detail would be helpful to explain the reasoning behind this hypothesis. SAMPLE SIZE: Pg 12 line 21: ***37 mmol/L is very different from 25 mmol/d. Why the shift from 37 mmol/L to 25 mmol/d in the power calculation? This should be explained. You could say that the target sample size will have 80% power to detect a change of 25 mmol/d but also give the specific power for detecting 37 mmol/L (or whatever the effect was in the Shandong study). DATA COLLECTION AND ANALYSIS ***In the analyses, will villages be nested within towns and towns nested within counties and counties within provinces? Is the nesting reflected appropriately within the calculations used to estimate sample size? The ways nesting is handled in the STATISTICAL ANALYSIS and SAMPLE SIZE sections are not clear. Pg 15 line 45: It is not clear how data collected from the systematic semi-structured interviews will be analyzed. It would be helpful to the reader if a sentence describing these analyses was added.
--	---

VERSION 2 – AUTHOR RESPONSE

Reviewer: 1

Reviewer Name: Christopher R. Long

Institution and Country: University of Arkansas for Medical Sciences

Please state any competing interests or state 'None declared': None declared

Please leave your comments for the authors below

This is an ambitious and important project, and it is clear that the authors have spent time carefully

revising this protocol based on reviewer comments. The introduction is well done--the explanation about the four RCTs is particularly helpful. There are still some minor areas where grammar and usage could be improved--I have listed a few in my comments, but I did not attempt to capture every sentence that could be improved. There a small number of areas that require more clarification. These major concerns are marked with three asterisks (***) . I look forward to reading about the results of this important and potentially influential study.

ABSTRACT:

1. "house cook" could be "home cooks"

Response: Thanks. We have corrected "house cook" to "home cooks".

2. "The secondary outcomes is the..." should read as the "The secondary outcomes are the ..."

Response: Sorry for our carelessness. We have corrected the error sentence "The secondary outcomes is the..." to "The secondary outcomes are the...".

STRENGTHS & LIMITATIONS:

3. In point #4, "and understand" could be "to understand"

Response: We have used "to understand" to replace "and understand".

4. Point #5 should be be divided into two sentences. Also "under the same county" could be "within the same county."

Response: We have divided into two sentences in point #5, and we corrected the word "under" to "within".

INTRODUCTION:

5. Pg 6 Line 29: The abbreviation LMIC is not used in the remainder of the manuscript and therefore is not necessary to introduce.

Response: Thanks for your advice and we deleted the abbreviation LMIC in the Line 29 of Pg 6.

METHODS AND ANALYSIS:

6. Pg 7 line 46: "The clusters are 48 towns (named street in urban areas, ... " could be clarified to "The clusters are 48 towns (called "streets" in urban areas, ... "

Response: The sentence "The clusters are 48 towns (named street in urban areas, ...)" have been corrected to "The clusters are 48 towns (called "streets" in urban areas, ...)"

7. Pg 7 line 52: In the phrase "where the livings of local residents", the word livings is confusing and could be replaced with a different word.

Response: We have used the word "lives" to replace "livings".

8. Pg 8 line 17: "the result of randomization keeps concealed " could be "the result of randomization is concealed"

Response: We have corrected "the result of randomization keeps concealed" to "the result of randomization is concealed".

9. Pg 8 line 26: "To avoid contamination," may be more precisely stated as "To minimize contamination,"

Response: We have used the word "minimize" to replace "avoid".

10. Pg 8 line 26: To minimize the introduction of multiple terms describing the same concept, is this "township or the cluster level" more precisely described as "the town level"?

Response: Many thanks, we have corrected "township or the cluster level" to "the town level".

11. Pg 9 line 21: "The reasons why not eligible for some residents" could be "The reasons why specific residents are ineligible"

Response: We corrected "The reasons why not eligible for some residents" to "The reasons why specific residents are ineligible".

INTERVENTION:

12. ***If people are randomized at the town level, but interventions will be carried out by county level staff, it is mentioned that any county level interventions could contaminate the control variables. Are all intervention activities implemented at the town or village level? It is mentioned under the "Salt reduction publicity section" but this perhaps should be emphasized clearly in the beginning of the intervention section.

Response: This has been stated in Study Setting and Overall Design section under METHODS AND ANALYSIS, and also re-emphasized in the first paragraph of Intervention as "The county CDCs will lead to deliver the intervention at township level, including mass publicity and education, interventions by communities, schools and catering units, and salt reduction interventions based on primary care institutions. Potential contamination may exist because the intervention is led by investigators at county level and residents in control group may visit people or eat at restaurants in intervention towns. Not adjacent to each other and restricting intervention within intervention towns will minimize the contaminations."

13. Pg 9 line 35: The term "propaganda centers" is likely to have a generally negative connotation for many English-speaking non-Chinese readers. There may be a more precise term to describe these locations for readers not from China.

Response: Thanks, we have corrected "propaganda centers" to "publicity department".

14. Throughout the manuscript, whenever contamination is mentioned, it may be more accurate to say that contamination will be "minimized" rather than "avoided"

Response: We have corrected the word "avoided" to "minimized".

15. Pg 10 line 36: It is not clear what is meant by "salt limiting spoons"

Response: We explained the salt limiting spoon in the manuscript. That is a plastic spoon specially designed to hold salt during cooking. The salt per spoon is 2g, which is convenient for home cooks to count and control the salt used during cooking.

16. Pg 11 line 16: Does "restaurants with a certain scale" mean "restaurants who serve a large enough number of clients," "restaurants who have a large enough number of locations," or something else?

Response: We did have concern previously that very small restaurants might be easily influenced by environment and the fidelity to the intervention is bad. So we hope to deliver the intervention in restaurants with "certain scale" (the capacity of receiving consumers is not too small). But this seems not to be the case. We have removed the restriction of "with a certain scale" and rewrite the paragraph as "Salt reduction activities can be carried out in restaurants, and canteens at workplaces located in the intervention towns. The knowledge of salt reduction can be publicized through posters, table decorations, and accessible brochures, so as to create a restaurant environment conducive to salt reduction. In the restaurants and the canteens, the catering chiefs are provided with standardized training at least once a year on how to reduce the use of salt and salty sauces during cooking. Consumers are encouraged to order food with less salt by waiters/waitresses."

17. Pg 12 line 10: ***Why is the hypothesis that the national level scale up will have the same impact

across both groups? A little more detail would be helpful to explain the reasoning behind this hypothesis.

Response: In the second year, we will try to equally deliver a national scale-up through the national CDC system, with no differentiate input or support among different areas, so we assume that the effects are the same under the interventions. This has been clarified in the context.

SAMPLE SIZE:

18. Pg 12 line 21: ***37 mmol/L is very different from 25 mmol/d. Why the shift form 37 mmol/L to 25 mmol/d in the power calculation? This should be explained. You could say that the target sample size will have 80% power to detect a change of 25 mmol/d but also give the specific power for detecting 37 mmol/L (or whatever the effect was in the Shandong study).

Response: We have added an explanation in our manuscript. Here is a copy: "Our recently completed Shandong salt reduction project with similar comprehensive intervention showed that the before-after effectiveness of sodium reduction was 37 mmol/d . Considering the before-after design may overestimate the effectiveness of intervention, we expect that our study will reduce sodium intake by at least 25 mmol/d (1.46 g/d salt) from baseline with comparison to the control group." If the estimated difference is 37 mmol/d, the power will be 99.1%, much higher than 80%. As the 37 mmol/d is an overestimation, we did not give the exact power to it in the text. Hope you are satisfied with this.

DATA COLLECTION AND ANALYSIS

19. ***In the analyses, will villages be nested within towns and towns nested within counties and counties within provinces? Is the nesting reflected appropriately within the calculations used to estimate sample size? The ways nesting is handled in the STATISTICAL ANALYSIS and SAMPLE SIZE sections are not clear.

Response: Thanks for your careful consideration and suggestion. We now have clarify that the intraclass correlation coefficient (ICC) of 0.080 is at village level in the part of SAMPLE SIZE section. In the STATISTICAL ANALYSIS part, we have a further discussion with our statistician and rewrote this part clarifying: (1) "with participants nested within village units and villages nested within towns" (No other above administrative levels considered because the randomization and intervention is at township level), and (2) the 12 months and 24 months effectiveness will be estimated separately using the same way without considering repeated measurement. Please see the revised manuscript for detail.

20. Pg 15 line 45: It is not clear how data collected from the systematic semi-structured interviews will be analyzed. It would be helpful to the reader if a sentence describing these analyses was added.

Response: Yes, it is. We have rewritten it as below:

At the end of year 1 and year 2, a systematic semi-structured interviews will be conducted separately to evaluate the fidelity and acceptability of each components of intervention. It will adopt an approach consistent with the UK MRC Guidelines for process evaluations of complex intervention . It will enable us to answer whether the intervention is effective, and what are the barriers and enablers for the potential scale-up. This will be carried out through a combination of in depth interviews with participants, intervention staff at county, town and village level, and policy makers, as well as interrogation of data collected during the trial.

REFERENCES

Chen X, Guo X, Ma J, et al. Urinary sodium or potassium excretion and blood pressure in adults of Shandong province, China: preliminary results of the SMASH project. *J Am Soc Hypertens* 2015, 9(10): 754-762.

Moore GF, Audrey S, Barker M, et al. Process evaluation of complex interventions: Medical Research Council guidance. *BMJ*, 2015, 350:h1258.

REVIEWER	Christopher Long University of Arkansas for Medical Sciences, USA
REVIEW RETURNED	27-Nov-2019

GENERAL COMMENTS	This manuscript describes an ambitious and potentially important public health project. The findings from this project—including findings related to participant outcomes and the process of implementing the project—have the potential to make a meaningful contribution to the field of sodium reduction interventions. This new revision has improved clarity over the initial version. The authors have addressed the specific comments that I made on the previous version of the manuscript. However, I am not sure that they have fully addressed either of the concerns raised by the editor in her comments. The strengths and limitations section lists characteristics of the study methods, but that section does not explicitly address strengths or weakness as such. (For example, a limitation of the study may be that there is no third group that receives neither the Year 1 local intervention nor the Year 2 national intervention. It may therefore be complicated to fully disentangle the effects of the Year 1 and Year 2 interventions.) Also, there remain several areas where typical English usage is not followed. For example, I have pasted a paragraph below where data collection activities are described as taking place in the future, the past, and the present in three consecutive sentences, respectively. Over the course of the revisions, the manuscript has improved with respect to describing the study activities. Most of the difficulties with readability do not greatly obscure the description of the study activities but instead make some of the sentences hard to follow (e.g., with respect to subject-verb agreement or verb tense). Example paragraph (page 13): “The questionnaire survey will be conducted a face-to-face by trained and qualified investigators through a mobile electronic data collection system. According to the content and sequence of the questionnaire, the basic information of the respondents and relevant behavioral risk factors, knowledge, attitudes and behaviors related to salt reduction and prevention and control of hypertension, hypertension and related expenses were collected. Physical measurements are accurately measured by trained researchers using calibrated measuring instruments, including height, weight, waist circumference, blood pressure, and heart rate.”
---

VERSION 3 – AUTHOR RESPONSE

Reviewer: 1

Reviewer Name: Christopher Long

Institution and Country: University of Arkansas for Medical Sciences, USA

Please state any competing interests or state ‘None declared’: None declared

Please leave your comments for the authors below

This manuscript describes an ambitious and potentially important public health project. The findings from this project—including findings related to participant outcomes and the process of implementing the project—have the potential to make a meaningful contribution to the field of sodium reduction interventions. This new revision has improved clarity over the initial version. The authors have

addressed the specific comments that I made on the previous version of the manuscript. However, I am not sure that they have fully addressed either of the concerns raised by the editor in her comments. The strengths and limitations section lists characteristics of the study methods, but that section does not explicitly address strengths or weakness as such. (For example, a limitation of the study may be that there is no third group that receives neither the Year 1 local intervention nor the Year 2 national intervention. It may therefore be complicated to fully disentangle the effects of the Year 1 and Year 2 interventions.) Also, there remain several areas where typical English usage is not followed. For example, I have pasted a paragraph below where data collection activities are described as taking place in the future, the past, and the present in three consecutive sentences, respectively. Over the course of the revisions, the manuscript has improved with respect to describing the study activities. Most of the difficulties with readability do not greatly obscure the description of the study activities but instead make some of the sentences hard to follow (e.g., with respect to subject-verb agreement or verb tense).

Example paragraph (page 13):

“The questionnaire survey will be conducted a face-to-face by trained and qualified investigators through a mobile electronic data collection system. According to the content and sequence of the questionnaire, the basic information of the respondents and relevant behavioral risk factors, knowledge, attitudes and behaviors related to salt reduction and prevention and control of hypertension, hypertension and related expenses were collected. Physical measurements are accurately measured by trained researchers using calibrated measuring instruments, including height, weight, waist circumference, blood pressure, and heart rate.”

Response: Thanks for your suggestions. We have rewritten the part of Strengths and limitations of this study. And we used professional language editing service (American Journal Experts) to revise the article. Hope you are satisfied.